# Urban density and depression during COVID-19 in Seoul: Moderating effects of social participation

Junseong Park[1], Ja-Hoon Koo[1], Sungik Kang[2]*

**1** Department of Urban and Regional Development, Hanyang University, Seoul, South Korea, **2** Institute of Urban Sciences, University of Seoul, Seoul, South Korea

\* namugnel@gmail.com

## Abstract

The COVID-19 pandemic, a global public health crisis, has had a profound and long-lasting impact on mental health worldwide. While existing studies on the pandemic have examined the relationship between urban characteristics and mental health, there remains a lack of understanding of the temporal distinctions of the pandemic, age-specific differences, and the role of social participation. This study empirically analyzed the relationship between COVID-19-related depressive symptoms and urban density characteristics, with a focus on the moderating effect of social participation by age group and pandemic phase. Using data from 25 Seoul districts in 2020 (initial phase) and 2021 (escalation phase), we measured depressive symptoms using a 10-point self-report scale. Urban density characteristics—population mobility density, residential density, and public transportation congestion—were derived from mobile phone signal data and building registries. We conducted ordinal logistic regression analyses with separate models for three age groups to examine associations between urban density, social participation, and depressive symptoms. First, the results revealed that population mobility density, residential density, and public transportation congestion were consistently positively associated with depressive symptoms across both study periods (2020 and 2021). Second, social participation significantly moderated the relationship between urban density characteristics and depressive symptoms, regardless of the pandemic phase. Third, the moderating effect of social participation on the relationship between urban density and depressive symptoms varied by age group. Specifically, these interaction effects were more pronounced among older adults. This study highlights the importance of understanding the complex relationships among urban density, social participation, and depressive symptoms during the COVID-19 pandemic.

**Data availability statement:** Data are available from Seoul Metropolitan Government's Community Survey through formal application to Seoul Open Data Plaza (https://data.seoul.go.kr). The specific datasets used in this study are the "Seoul Survey 2020" and "Seoul Survey 2021" available at: https://data.seoul.go.kr/dataList/OA-15564/F/1/datasetView.do#.

**Funding:** This work was supported by the Korea National Research Foundation under Grant [RS-2023-00239319].

**Competing interests:** The authors have declared that no competing interests exist.

## Introduction

The novel coronavirus disease (COVID-19) emerged as a new infectious disease in December 2019, posing a significant threat to the physical and mental health of the global population through symptoms such as fever, chills, and muscle aches. COVID-19, which initially spread across Asian countries during January and February 2020, was rapidly transmitted worldwide from March onwards and was declared a pandemic, the highest warning level for infectious diseases, by the World Health Organization (WHO) on March 11, 2020. The COVID-19 pandemic has profoundly affected the lives of millions of people worldwide by altering their ways of living, working, and interacting with others. According to COVID-19 statistics provided by the WHO, as of April 2024, the number of confirmed cases globally exceeded 700 million, with approximately 7 million cumulative deaths reported [1]. In response to the spread of the pandemic, governments worldwide implemented various non-pharmaceutical interventions, including restrictions on physical and social interactions and mobility. These measures can be broadly categorized into individual preventive behaviors and government-level policies. At the individual level, practices such as wearing masks, maintaining hand hygiene, and social distancing were recommended and found effective in reducing virus transmission [2,3]. At the government level, policies such as border and regional lockdowns, bans on gatherings, and recommendations for remote work were implemented and evaluated as effective in slowing the spread of the infection [4,5]. Despite these efforts, anxiety about infection, feelings of isolation, and depressive symptoms emerged as significant social challenges that threaten individuals worldwide [6].

The non-pharmaceutical preventive measures, coupled with pandemic-related stressors, exposed individuals to extensive risk factors for mental health issues, such as worry, anxiety, and feelings of isolation. Consequently, significant mental health problems, particularly depressive symptoms, have been exacerbated worldwide [7,8]. One of the primary drivers of mental health deterioration has been identified as the fear and anxiety of potentially transmitting the virus to family, friends, or oneself. Several studies have highlighted a strong correlation between COVID-19-related fear and depressive symptoms, anxiety, and stress, observed across all age groups [9]. Notably, this anxiety about becoming a potential "spreader" of infection has been pronounced not only among the general public and workers but also among healthcare professionals, leading to heightened levels of stress and burnout. These fears and anxieties intensify social isolation and restrict interpersonal relationships, thereby increasing the risk of long-term mental health problems [10,11]. These findings underscore the importance of addressing fear, anxiety, and depressive symptoms related to infection during a pandemic and highlight the need to incorporate these considerations into public health policies aimed at maintaining the mental health of individuals and communities.

South Korea experienced similar challenges during the pandemic, with the government implementing various social measures to minimize the impact of COVID-19 and respond to the resulting socioeconomic changes. In early 2020, when COVID-19

first emerged, several preventive measures were implemented. These included restrictions on social contact (e.g., limiting face-to-face interactions), workplace-related contact restrictions (e.g., closure of shops and restaurants, encouragement of remote work), limitations on daily activities (e.g., reduction of leisure activities in public spaces), closure of educational institutions (e.g., schools, universities, and kindergartens), and border closures [12]. These preventive measures caused significant disruptions to the economic system and personal lives of individuals, leading to issues such as job losses and income reduction [13]. The expansion of remote work and online education substantially increased reliance on digital technologies, accelerating the rapid spread of "untact" culture [14,15]. Social distancing measures restricted in-person gatherings and leisure activities, resulting in the contraction of individuals' social networks and an increase in cases that negatively affected their mental health. Additionally, industries dependent on face-to-face services, such as the restaurant, travel, and performance sectors, suffered significant economic damage. This economic downturn led to increased job insecurity and widened income disparities, further exacerbating socio-economic inequality [16].

This study explored the relationship between COVID-19-related depressive symptoms, urban density characteristics, and social participation in South Korea based on the significance of these issues. While existing research on the COVID-19 pandemic has examined the impact of urban density on mental health, there remains a limited understanding of the differences between 2020 and 2021 and across age groups. To address this gap, this study focused on urban density characteristics and social participation during the pandemic, investigating how depressive symptoms related to COVID-19 varied by pandemic phase and age group. To achieve this objective, Chapter 2 reviews previous studies that examined the relationship between mental health and urban density during the pandemic and formulates research questions based on their findings. Chapter 3 describes the dataset, variable measurement methods, and analytical approach used to examine the relationship between COVID-19-related depressive symptoms and urban density characteristics. Chapter 4 presents the results of the empirical analysis and discusses their implications. Finally, Chapter 5 summarizes the study's content and findings and addresses its limitations.

## Literature review

The COVID-19 pandemic has had a widespread negative impact on physical and mental health worldwide. The direct risk of infection, the fear associated with it, and the economic and social changes resulting from preventive measures have significantly influenced mental health, as confirmed by various studies. A meta-analysis indicated that anxiety symptoms increased by approximately 25% globally during the first year of the pandemic [17], attributing to factors such as social isolation, economic stress, and health and safety concerns. Furthermore, compared to the anxiety prevalence rate of approximately 6% in some regions in 2017, studies reported that during the pandemic, this rate exceeded 19% in certain population groups [18]. During the COVID-19 outbreak in April 2020, a study conducted among U.S. adults revealed that 13.6% of respondents reported severe psychological distress, a significant increase from the 3.9% reported in 2018 [19]. These findings underscore the changes in psychological symptoms caused by COVID-19. Notably, among the fear, worry, and anxiety related to the pandemic, the fear of oneself or family members contracting the virus emerged as one of the most significant factors [20].

This unprecedented crisis has had a significant impact on mental health across age groups. Among middle-aged adults, particularly parents with young children, pandemic-related stressors interacted in complex ways. School closures and reduced opportunities for peer interaction contributed to increased screen time and sleep disturbances among children, significantly heightening parental stress [21–23]. Financial concerns were linked to higher verbal aggression, while loneliness was associated with child neglect, and worry increased the risk of physical abuse [24]. Older adults faced greater health risks and were more likely to experience social isolation from family and social connections, increasing their vulnerability to mental health symptoms, cognitive decline, and impairment in daily functioning [25,26]. European studies revealed that individuals aged 60 and older exhibited higher tendencies to worry about COVID-19 compared to younger

age groups [27]. Thus, while middle-aged adults experienced complex parenting-related stressors, older adults were particularly vulnerable due to infection risks and social isolation.

Meanwhile, as the effectiveness of preventive measures increased and infection and mortality rates decreased, mental health issues such as depression, fear, and anxiety tended to recover. Studies indicated that mental health analyses before and after the COVID-19 pandemic revealed the most pronounced increase in depression during the first two months following the pandemic declaration (April 2020), followed by a return to pre-pandemic levels by mid-2020 [28]. Additionally, a study conducted in China found that the levels of stress, anxiety, and depression observed during the initial outbreak period (January–February 2020) did not worsen even four weeks after the epidemic reached its peak. These findings suggest that individuals may have adapted to the new circumstances after the initial phase [29]. According to mental health assessments in the United Kingdom, clinically significant mental health morbidity increased from 19% in 2018–2019 to 27% in April 2020 (one month after the UK lockdown). However, most individuals subsequently recovered, reporting consistently good (39%) or very good (38%) mental health [30]. Furthermore, the recovery group (12%) experienced a deterioration in mental health due to the initial shock of the pandemic but later returned to pre-pandemic mental health levels [31]. These findings indicate that while depression due to COVID-19 surged dramatically during the early stages of the pandemic, it gradually decreased over time. This recovery can be attributed to individuals' adaptation, the effectiveness of preventive measures, a reduction in actual risks, and medical advancements such as vaccines and treatments, which reduced the risk of death and alleviated psychological distress.

With the recent global trend of urbanization, academic interest in the impact of urban environments on mental health has grown. Urban environments encompass a variety of factors that have complex influences on mental health, often skewed toward negative effects. Even after controlling for individual characteristics, urban environmental factors such as access to natural environments, proximity to facilities, architectural features, housing density, and housing quality significantly affect mental health [32]. Better housing quality, less crowded neighborhoods, lower building density, and greater accessibility to facilities were associated with positive effects on mental health. Moreover, a study focusing on New York City revealed that individuals living in high-density housing areas were more likely to experience depression [33]. These findings suggest that environmental stressors such as high population density, noise, and air pollution in urban centers, as well as the limited personal space and disconnection from nature associated with apartment living, may negatively impact mental health [34].

Notably, the relationship between urban density and mental health became more pronounced during infectious disease outbreaks. High population density and complex social networks in urban areas not only accelerated the spread of infectious diseases but also amplified residents' anxiety and fear of infection. Multiple studies have documented heightened mental health risks in urban areas during COVID-19. Research across the United States, Belgium, and the Netherlands consistently found that urban residents reported higher levels of depressive symptoms and anxiety compared to rural residents, with effects being particularly pronounced in densely populated areas and large cities with high infection rates [35–37]. Urban density characteristics such as high population density, significant mobility, housing instability, and low socioeconomic status were associated with higher prevalence rates of depression and anxiety disorders [38–41]. These findings suggest that urban environments, particularly high-density settings, may pose elevated risks for mental health problems during infectious disease outbreaks.

Numerous studies have revealed the dual impact of urban environments on mental health. In areas with well-developed public transportation infrastructure and urban amenities, residents tended to experience higher life satisfaction and lower psychological distress. However, particularly during infectious disease outbreaks, urbanized areas—characterized by high population density, significant mobility, and numerous public gathering places—faced increased risks of disease transmission and heightened risks of mental health deterioration as a consequence [12,36]. These findings consistently demonstrate that urban characteristics, particularly high population densities and the associated risk of infectious disease spread, can significantly affect the mental health of residents. Therefore, to establish effective mental health

management policies suited to urban environments, further research is needed to deepen our understanding of how the spread of infectious diseases such as COVID-19 affects mental health in high-density urban settings [37].

This study aimed to explore the relationship between urban factors contributing to the spread of infections and depressive symptoms during the COVID-19 pandemic and the role of social participation in this relationship (Fig 1). While prior research has established associations between urban density and mental health during COVID-19, three critical gaps remain. First, although some studies have examined temporal changes in general mental health [28–31], most urban density and mental health studies during COVID-19 used single-timepoint cross-sectional designs [35–41], precluding analysis of how urban density-depression relationships evolved across different pandemic phases. Second, while age differences in COVID-19 vulnerability and mental health are well documented [21–27,32], existing urban density studies have not examined whether urban density effects on depression vary across age groups [35–41]. Third, although social participation protects mental health during crises [42–44], its potential to buffer urban density stressors during infectious disease outbreaks remains unexplored.

Understanding these dynamics is essential for developing age-specific and phase-appropriate urban mental health interventions during pandemics. Therefore, the present study examined the relationship between urban density characteristics and COVID-19-induced depressive symptoms while considering temporal and age-specific variations, and analyzed the moderating role of social participation. Against this backdrop, this study poses the following questions: 1) Which urban density characteristics were associated with an increase in depressive symptoms caused by COVID-19? 2) Does social participation effectively mitigate the increase in depressive symptoms associated with urban density? 3) Do the moderating effects of social participation vary according to the pandemic phase and age group?

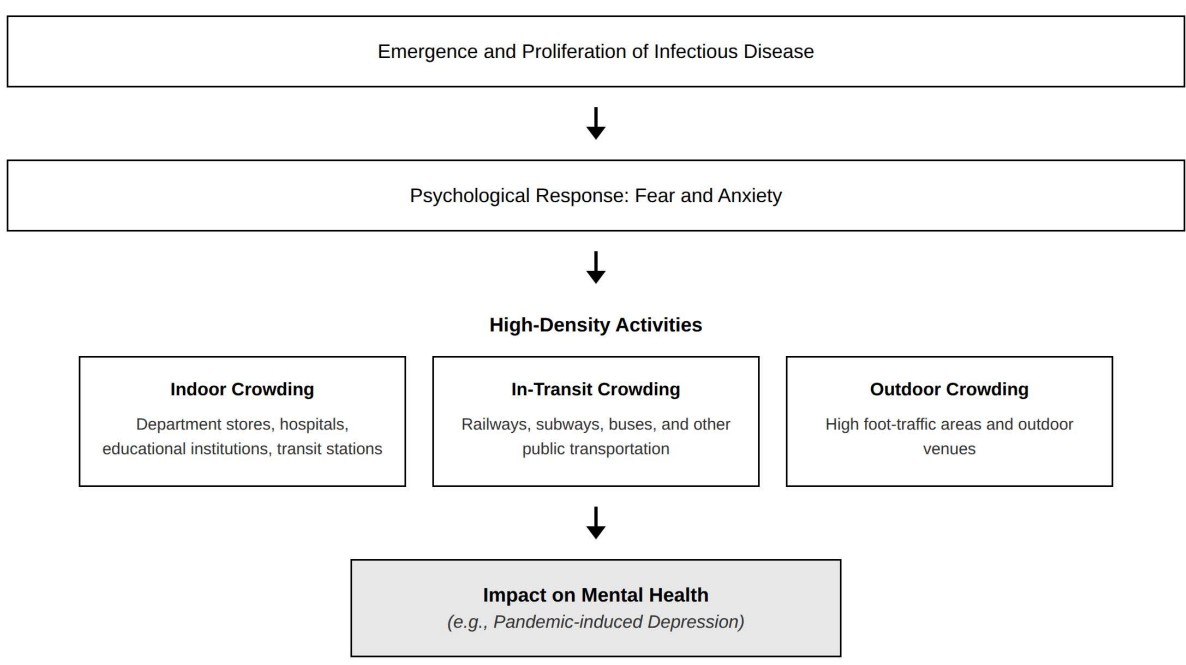

**Fig 1. Urban density and its psychological impact during infectious disease outbreaks.**

## Materials and methods

### COVID-19 outbreak and characteristics in Seoul

This study analyzed the COVID-19 pandemic in Seoul by dividing it into two periods: 2020 and 2021. This classification was based on patterns of mental health changes and the timeline for the recovery of daily living activities reported in previous studies. 2020 included the first and second waves, whereas 2021 included the third and fourth waves (Table 1; Fig 2). The first wave in 2020 spanned from January 20, 2020, to August 11, 2020, and was characterized by cases imported from abroad and outbreaks concentrated in the Daegu, Gyeongbuk, and Seoul metropolitan areas. During this period, the first confirmed cases emerged from international arrivals, followed by a significant outbreak in the Daegu and Gyeongbuk regions associated with a specific religious group. On June 28, 2020, the government introduced a social distancing system. During the first wave, the daily average number of confirmed cases was 71.5, with 375 severe cases and 308 deaths. The second wave in 2020 lasted from August 12, 2020, to November 12, 2020, and was marked by the spread of COVID-19 in the Seoul metropolitan area. The outbreak was driven primarily by religious facilities and large-scale urban protests in the region. On November 1, 2020, the government implemented a five-level social distancing system. During the second wave, the daily average number of confirmed cases rose to 142.8—approximately double that of the first wave—with a total of 575 severe cases and 221 deaths.

The third wave of the escalation phase was defined as the nationwide spread period from November 13, 2020, to July 6, 2021. During this period, variants such as Alpha and Delta strains emerged and drove the spread of the virus, shifting the focus from a metropolitan-centered outbreak to nationwide transmission. Notable events during this time included the initiation of COVID-19 vaccinations on February 26, 2021, and the implementation of a revamped four-tier social distancing system on July 1, 2021. The fourth wave, which lasted from July 7, 2021, to January 29, 2022, was characterized by the spread of the Delta variant. As the Delta variant became the dominant strain, confirmed cases continued to emerge across various facilities. The daily average number of confirmed cases during this wave surged to 3,137.8, a significant increase compared to previous waves. However, the fatality rate decreased to less than half that observed during the first wave. Key events during this period included achieving a 70% first-dose vaccination rate among the population on

**Table 1. Characteristics of COVID-19 Pandemic Phases.**

| Category | Initial phase(2020) | | Escalation phase(2021) | | Recovery phase |
|---|---|---|---|---|---|
| | First wave | Second wave | Third wave | Fourth wave | Fifth wave |
| | 2020.01.20-2020.08.11 | 2020.08.12-2020.11.12 | 2020.11.13-2021.07.06 | 2021.07.07-2022.01.29 | 2022.01.30-2022.04.24 |
| Key Characteristics | Overseas and Daegu/Gyeongbuk-Metropolitan Area Outbreak | Metropolitan Area Spread | Metropolitan & Nationwide Spread | Delta Variant Spread | Omicron Variant Spread |
| Daily Avg. Confirmed Cases (Min~Max) | 71 (1~909) | 142 (38~441) | 566 (191~1,240) | 3,137 (1,049~17,509) | 187,424 (17,075~621,177) |
| Severe Cases (Daily Avg.) | 375 (2) | 575 (6) | 3,188 (13) | 9,130 (44) | 8,869 (103) |
| Deaths (Fatality Rate) | 308 (2.1%) | 221 (1.6%) | 1,556 (1.1%) | 5,061 (0.7%) | 15,899 (0.1%) |
| Reproduction Number | 0.53~9.35 | 0.68~3.05 | 0.79~1.52 | | |
| Government Policies | Social Distancing Initiated (2020.06.28) | Five-Level Social Distancing Initiated (2020.11.01) | Vaccination Began (2021.02.26); Revised to Four-Level Social Distancing (2021.07.01) | 70% First Dose Vaccination Achieved (2021.09.17); Gradual Transition to Normalcy (2021.11.01) | Social Distancing Lifted (2022.04.28) |
| Survey Period | 2020.08 | | 2021.08 | | – |

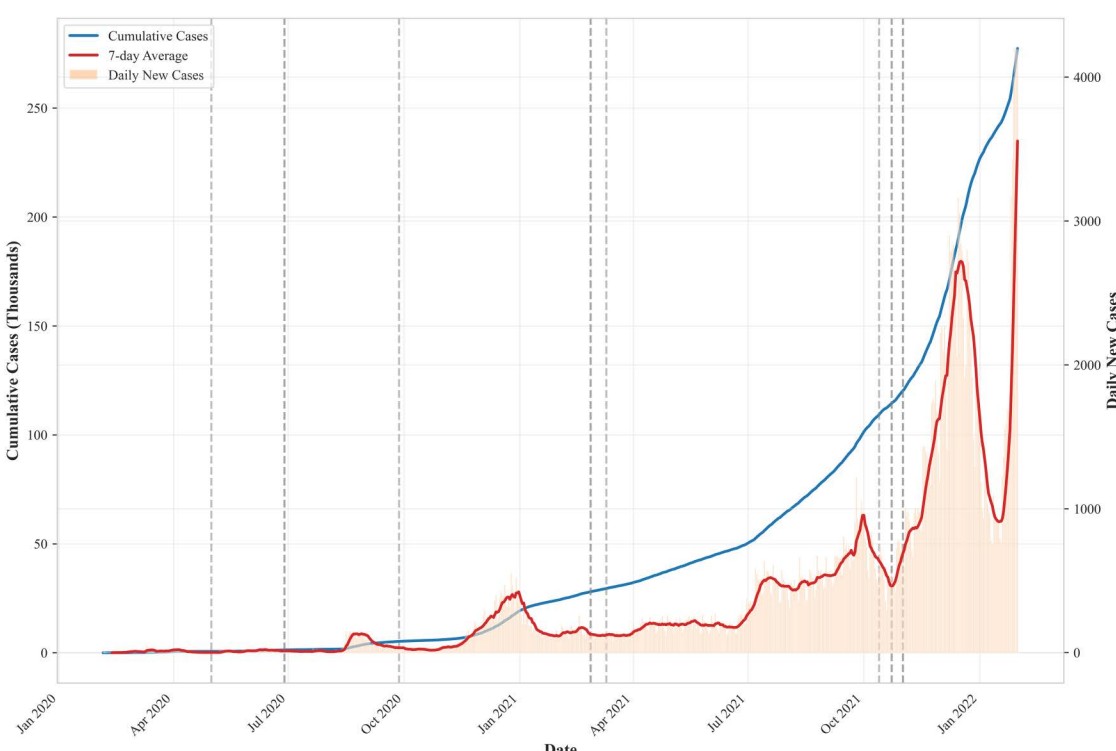

**Fig 2. COVID-19 Outbreak Status in Seoul.**

September 17, 2021, and the phased implementation of a return to normalcy starting November 1, 2021. This study excluded the fifth wave (January 30, 2022, to April 24, 2022) from the analysis. Although this period, driven by the Omicron variant, accounted for approximately 95% of the total cumulative cases, it coincided with the transition to 2022, making it less relevant to the analytical focus of this study. By distinguishing between these periods, this study focused on 2020 and 2021, during which the social impacts of COVID-19 were most pronounced.

## Data and measurements

The data used in this empirical analysis were drawn from the Citizen Survey of the Seoul Survey on Urban Policy Indicators, conducted by the Seoul Metropolitan Government. Data were collected through in-person household interviews by trained surveyors or, in cases where interviews were declined due to COVID-19, through online non-contact surveys [45]. The Seoul Citizen Survey annually targeted 5,000 individuals aged 15 years and older residing in Seoul, yielding a total sample of 10,000 participants across the two study periods (2020: n = 5,000; 2021: n = 5,000). The sample was selected using a stratified cluster sampling method, dividing the population into multiple strata based on the demographic and regional characteristics of Seoul. The primary objective of the Seoul Citizen Survey is to collect comprehensive data on various areas directly related to Seoul residents' quality of life. The survey covered a wide range of topics, including population and household composition, economic activity, housing conditions, health status, perceptions of safety, environmental issues, transportation use, cultural activities, education, and citizen values. Notably, the survey incorporated additional COVID-related questions during the pandemic to assess changes in citizens' perceptions during this period. Given the survey's ability to facilitate various analyses related to COVID-19, this study used data collected from the Seoul Citizen

Survey. Data from the early phase of the pandemic were collected between September 14 and October 31, 2020, while data from the escalation phase were collected between September 6 and November 8, 2021.

The dependent variable was depressive symptoms during the COVID-19 pandemic. In the 2020 and 2021 Seoul Citizen Survey, new questions related to COVID-19 were introduced, including items assessing the degree of depression caused by COVID-19 and the main factors contributing to feelings of depression during the pandemic. Depressive symptoms related to COVID-19 was measured using the question, "Since the onset of COVID-19, have you ever felt depressed in your daily life due to the pandemic?" Survey participants rated their responses on a scale ranging from 0 (not at all depressed) to 10 (extremely depressed).

Although this single-item measure differs from multidimensional scales such as the PHQ-9 or CES-D, single-item assessments of depressive symptoms have demonstrated validity in large-scale epidemiological studies. For example, Zimmerman et al. (2006) reported that a single-item measure of depression symptom severity showed a strong correlation with the 18-item Clinically Useful Depression Outcome Scale (CUDOS) (r = 0.78, p < 0.001) and a moderate correlation with the clinician-rated Clinical Global Impression-Severity scale (CGI-S) (r = 0.64, p < 0.001) among 562 outpatients [46]. Van Rijsbergen et al. (2014) reported that a single-item mood scale demonstrated 94% discriminative accuracy (AUC = 0.94) against the Structured Clinical Interview for DSM-IV (SCID-I) conducted by psychiatric professionals [47]. Mallon et al. (2002) demonstrated 81% agreement between a single depression item and the 14-item Hospital Anxiety and Depression Scale (HADS) in a large epidemiological study [48].

The single-item format was adopted by the Seoul Metropolitan Government to enable rapid capture of mental health impacts attributable to COVID-19 while minimizing respondent burden during the pandemic (N = 5,000 per wave). Our COVID-19-related depressive symptom measure assesses self-perceived depressive symptoms arising from the pandemic's direct effects rather than general depressive symptoms, focusing on monitoring COVID-19-related mental health changes. Similarly, Khajuria et al. (2021) demonstrated the effectiveness of a single depression item for rapid mental health assessment in pandemic contexts among 2,527 healthcare workers across 41 countries [49].

Based on previous studies examining the causes of the spread of urban infectious diseases [50–52], this study used population mobility density, residential density, and public transportation congestion as key urban density characteristics. Population mobility is an indicator of the volume of movement within a specific spatial and temporal range and is defined as the number of people traveling to and from a particular area over a given period. Population mobility was measured using the "Seoul Living Population" dataset, which has been provided by the Seoul Metropolitan Government since 2017. This dataset captures the current and mobile populations in Seoul at the time of measurement. The Seoul Living Population data, derived from KT (Korea Telecom) LTE mobile phone signal communication data, offer high-frequency measurements at hourly intervals, enabling precise and detailed tracking of population dynamics within the city. The dataset distinguishes between nighttime residential locations and daytime activity locations through algorithmic classification based on continuous presence patterns and mobile signal data from Korean nationals and long-term foreign residents.

Population mobility density was calculated as the maximum daily number of mobile individuals per district divided by the area of the district (Daily Maximum Mobile Population/ District Area in km²). The maximum daily mobile population includes the total number of individuals entering Seoul from outside, moving within the same district, and moving between districts. We employed daily maximum values rather than averages to capture peak infection transmission risk during concentrated contact periods, when congestion in public spaces is highest. Areas with high population mobility density may have a relatively higher risk of transmission of respiratory infectious diseases, such as COVID-19, which can increase fear of infection and depressive symptoms.

Among the urban density variables, residential density is a critical factor in the urban spatial structure. In this study, residential density is defined as the ratio of the total floor area of residential buildings to the total floor area of all buildings within each district, based on building registry data from the Public Data Portal. Areas with high residential density indicate a higher population density, which is likely to increase further during the COVID-19 pandemic owing to remote work

 

practices. Additionally, confirmed cases in these areas may be perceived as an immediate threat by residents, potentially increasing their anxiety and stress levels. Therefore, residential density serves as an essential indicator for understanding lifestyle patterns, social interactions, and subsequent mental health conditions of residents in different areas during the pandemic. Public transportation congestion, another key characteristic of urban density, was included in this study as a significant indicator of urban mobility patterns and perceptions of infection risk during the COVID-19 pandemic. This variable was calculated based on the daily average number of public transportation users in Seoul, defined as the sum of subway and bus ridership. Closely related to urban mobility, this indicator reflects the effectiveness of social distancing policies, changes in mobility patterns, and their association with COVID-19-related depressive symptoms during the pandemic.

The social participation variable was measured using the question, "In the past year, have you participated in any of the following meetings or group activities?" These group activities included community activities, such as social organization participation, hobby clubs, and local gatherings. The social participation variable was measured on a binary scale, with 0 indicating "no participation" and 1 indicating "participation." This variable serves as an indicator of an individual's level of social activity and plays a crucial role in exploring the relationship between social connectedness and depressive symptoms during the COVID-19 pandemic. Social participation is known to have a positive effect on mental health; however, this relationship may become more complex during a pandemic. Participation in social activities can provide social support and a sense of belonging, potentially reducing depressive symptoms. However, it may increase anxiety regarding the risk of infection, thereby exacerbating depressive symptoms. By analyzing this variable, this study aimed to investigate the complex effects of social participation on depressive symptoms during the pandemic and gain insights into the relationship between social distancing and mental health.

Individual characteristics that may influence COVID-19-related depressive symptoms were included in the analysis model as control variables, based on prior studies. Participants were categorized into three age groups for analysis: young adults (19–34 years), middle-aged adults (35–64 years), and older adults (65 years and above), with sample distributions detailed in Table 2. These characteristics included sex (male, female), age, educational level (high school graduate or less, associate degree or higher), cohabitation status (living alone or living with others), employment status (unemployed or employed), and housing type (apartment or non-apartment housing). Gender was analyzed using "male" as the reference category, and age was treated as a continuous variable based on the participant's self-reported age in years. Cohabitation status was treated as a categorical variable, with 0 indicating living alone and 1 indicating living with others. Employment status was measured as a binary variable and categorized as unemployed or employed. Education level used "high school graduate or less" as the reference category. Housing type was measured using apartments, the most common form of residence, as the reference category.

**Table 2. Analysis of Differences in COVID-19-related depressive symptoms by pandemic phase and age group.**

| Variable | Young adults | | Middle-aged adults | | Older adults | | ANOVA (F-test) |
|---|---|---|---|---|---|---|---|
| | Mean | S.D. | Mean | | S.D. | Mean | S.D. |
| **2020** | 6.04 | 2.13 | 6.08 | 1.96 | 6.49 | 1.66 | 20.19*** |
| | (N=1,227) | | (N=2,695) | | (N=1,080) | | |
| **2021** | 5.20 | 2.41 | 5.30 | 2.23 | 5.45 | 2.11 | 3.49* |
| | (N=1,302) | | (N=2,697) | | (N=1,001) | | |

S.D. = Standard deviation.

***p<0.01.

**p<0.05.

*p<0.1.

## Analytical methods

We used an Ordered Logistic Model (proportional odds model) to examine the relationship between urban density and COVID-19-related depressive symptoms. This regression analysis method was selected as the most appropriate, given that the dependent variable involved selecting a category from a range of 0 to 10. The binary logistic model calculates the probability of an event occurring ($Prob(Y = 1)$) and its complement ($1 - Prob(Y = 1)$), determines the ratio of these two probabilities, and transforms them into a natural logarithm to estimate the likelihood that Y is equal to 1. Similarly, the Ordered Logistic Model calculates cumulative probabilities for each rank when the dependent variable is measured as an ordinal variable ranging from 1 to n. This model uses cumulative odds, which represent the ratio of the probability of being at or below a particular category versus being above that category [$P(Y \leq j)/P(Y > j)$], under the proportional odds assumption that the coefficients remain constant across all cumulative logits. In logistic models, interpreting independent variables in terms of log odds is often more intuitive. The odds ratio (OR) reflects the extent to which changes in an independent variable affect the likelihood of an event occurring for the dependent variable while holding other variables constant. The results were interpreted based on these odds ratios. The regression equation for the Ordered Logistic Model used in this study is:

$$logit\left(\frac{Pr(Y_{ij})}{1-Pr(Y_{ij})}\right) = B_0 + B_1 Urban_{Density_j} + B_2 Social\_Participation_{ij}$$
$$+ B_3 Urban\_Density_j \cdot Social\_Participation_{ij} + B_4 X_{ij} + e_{ij}$$

## Results and discussion

### Results for descriptive analysis

Table 2 presents the sample distribution and differences in COVID-19-related depressive symptoms by pandemic phase and age group. In 2020, the sample comprised 1,227 young adults (1934 years), 2,695 middle-aged adults (35−64 years), and 1,080 older adults (≥65 years). In 2021, the distribution was 1,302 young adults, 2,697 middle-aged adults, and 1,001 older adults. The results of the ANOVA test showed significant differences in depressive symptoms caused by COVID-19 across both years and age groups (2020: 20.19, $p < 0.01$; 2021: 3.49, $p < 0.05$). During 2020, depressive symptoms are generally more frequent and tend to increase with age. Specifically, the average depressive symptom scores were 6.04 (±2.13) for young adults, 6.08 (±1.96) for middle-aged adults, and 6.49 (±1.66) for older adults. In 2021, depressive symptoms decreased overall; however, the trend of higher depressive symptoms among the older age groups persisted. The average scores were 5.20 (±2.41) for young adults, 5.30 (±2.23) for middle-aged adults, and 5.45 (±2.11) for older adults. Notably, depressive symptoms decreased in 2021 compared with 2020 across all age groups. This finding indicates a temporal decline in COVID-19-related depressive symptoms during the study period. Additionally, older adults consistently exhibited higher levels of depressive symptoms than other age groups.

Descriptive statistics for the urban density characteristics—population mobility, residential density, and public transportation congestion—in Seoul are presented in Table 3. The average population mobility was 302.1 thousand people (Standard Deviation (SD): 112.9 thousand people), indicating that, on average, 302.1 thousand people were mobile in the area daily. The minimum value was 179.9 thousand people, and the maximum was 843.1 thousand people. The average residential density across Seoul was 54.7% (SD: 11.8%), meaning that 54.7% of the total floor area of all the buildings in the area was used for residential purposes. The minimum and maximum values were 19.9% and 69.2%, respectively. For public transportation congestion, the average congestion level was 309.9 thousand people (SD: 120.0 thousand people), indicating that, on average, 309.9 thousand people used public transportation daily. The minimum value was 163.1 thousand people, whereas the maximum value was 728.8 thousand people, showing a difference of approximately 4.5 times.

**Table 3. Descriptive Statistics by COVID-19 Pandemic Phase.**

| Variable | Measurement | Initial phase(2020) | | | Escalation phase(2021) | | |
|---|---|---|---|---|---|---|---|
| | | Mean | S.D. | Range | Mean | S.D. | Range |
| **Dependent variable** | | | | | | | |
| **COVID-19-related depressive symptoms** | Experience of depressive feelings in daily life (0: Not at all depressed, 10: Extremely depressed) | 6.16 | 1.95 | 0-10 | 5.3 | 2.25 | 0-10 |
| **Independent variables** | | | | | | | |
| **Urban density feature** | | | | | | | |
| **Population mobility** | Daily maximum mobility/ district area (thousand people/km²) | 302.1 | 112.9 | 179.8-843.0 | 301.6 | 110 | 179.8-843.0 |
| **Residential density** | Residential floor area/ total floor area (km²) | 0.547 | 0.118 | 0.19-0.69 | 0.547 | 0.117 | 0.19-0.69 |
| **Public transportation congestion** | Daily average bus and subway rider-ship (thousand people) | 309.9 | 120 | 163.1-728.7 | 309.1 | 118.9 | 163.1-728.7 |
| **Social participation** | Participation in social activities such as local meetings and group activities (0: Not participating, 1: Participating) | 0.72 | 0.45 | 0-1 | 0.59 | 0.49 | 0-1 |
| **Sociodemographic Characteristics** | | | | | | | |
| **Gender** | 0: Male, 1: Female | 0.51 | 0.5 | 0-1 | 0.51 | 0.5 | 0-1 |
| **Age** | Continuous variable for age (18 years and older) | 48.34 | 16.15 | 18-92 | 47.72 | 16.03 | 18-94 |
| **Cohabitation Status** | 0: Living alone, 1: Living with others | 0.86 | 0.36 | 0-1 | 0.85 | 0.34 | 0-1 |
| **Employment Status** | 0: Unemployed, 1: Employed | 0.66 | 0.47 | 0-1 | 0.68 | 0.46 | 0-1 |
| **Education Level** | 0: High school graduate or less, 1: College graduate or higher | 0.61 | 0.48 | 0-1 | 0.63 | 0.48 | 0-1 |
| **Apartment Residency** | 0: Other, 1: Yes | 0.47 | 0.49 | 0-1 | 0.49 | 0.5 | 0-1 |

N = 5,000

S.D. = Standard deviation.

a: Data source is Seoul Open Data Plaza (https://data.seoul.go.kr).

Basic statistical analysis of the social participation variable, a social contact factor, yielded the following results. In 2020, 72% (SD = 0.45) of respondents reported engaging in social participation activities. However, this proportion decreased to 59% (SD = 0.49) in 2021. This suggests an overall decline in social participation as the pandemic progressed. This change can be interpreted as the combined result of social distancing policies, an increase in non-face-to-face activities, and heightened anxiety about infection during the spread of COVID-19. A reduction in social contact may be associated with an increase in depressive symptoms, making it an important indicator of the pandemic's impact on mental health. In addition, various sociodemographic characteristics that may influence COVID-19-related depressive symptoms were included as control variables. The control variables included sex (female: 51%), age (mean: 48 years), cohabitation status (cohabitation: 85%), employment status (employed: 67%), educational attainment (college graduate or higher: 62%), and housing type (apartment residence: 48%).

## Results of urban density characteristics

Table 4 presents the regression analysis results for COVID-19-related depressive symptoms during the early phase of the pandemic (2020), and Table 5 displays the results for the escalation phase (2021). The findings of this study revealed significant and consistent associations between urban density characteristics and depressive symptoms across different

Table 4. Results of COVID-19-Related Depressive Symptoms in 2020.

| | Model 1 (Young adults) | | Model 2 (Middle-aged adults) | | Model 3 (Older adults) | |
|---|---|---|---|---|---|---|
| | O.R. | S.E. | O.R. | S.E. | O.R. | S.E. |
| **Sociodemographic Characteristics** | | | | | | |
| **Gender (1 = female)** | 1.575*** | 0.161 | 1.301*** | 0.097 | 1.086 | 0.132 |
| **Age** | 1.005 | 0.017 | 0.995 | 0.005 | 1.031*** | 0.011 |
| **Cohabitation Status (1 = yes)** | 0.943 | 0.137 | 0.770** | 0.085 | 1.309* | 0.192 |
| **Employment Status (1 = yes)** | 0.915 | 0.124 | 0.918 | 0.085 | 1.018 | 0.133 |
| **Education Level (1 = College Graduate)** | 1.003 | 0.143 | 0.930 | 0.081 | 1.165 | 0.181 |
| **Marital Status (1 = yes)** | 0.761 | 0.141 | 1.297*** | 0.128 | 1.714 | 0.847 |
| **Apartment Residency (1 = yes)** | 0.866 | 0.094 | 0.842** | 0.060 | 0.880 | 0.099 |
| **Urban density feature** | | | | | | |
| **Population mobility ℗** | 1.207* | 0.134 | 1.151 | 0.103 | 1.577*** | 0.217 |
| **Residential density ®** | 1.207 | 0.142 | 1.207* | 0.120 | 1.470** | 0.256 |
| **Public transportation congestion ⓣ** | 0.941 | 0.087 | 1.138 | 0.094 | 0.917 | 0.152 |
| **Social participation** | 1.566*** | 0.167 | 1.147* | 0.091 | 1.136 | 0.165 |
| **Interaction term** | | | | | | |
| **Social participation * ℗** | 0.845 | 0.118 | 0.935 | 0.098 | 0.655** | 0.109 |
| **Social participation * ®** | 0.810 | 0.121 | 0.869 | 0.099 | 0.651** | 0.127 |
| **Social participation * ⓣ** | 1.072 | 0.123 | 0.912 | 0.085 | 1.016 | 0.181 |
| **N** | 1227 | | 2693 | | 1080 | |
| **Wald** | 44.00*** | | 49.21*** | | 30.52*** | |
| **AIC** | 5030.26 | | 10760.31 | | 3980.05 | |
| **BIC** | 5152.96 | | 10901.87 | | 4099.68 | |

O.R.=Odds Ratio, S.E.=Standard Error.

***p<0.01.

**p<0.05.

*p<0.1.

age groups in 2020 and 2021. Specifically, population mobility exhibited a positive association with depressive symptoms among older adults (Model 3: O.R. = 1.577, p<0.01), and this association persisted in 2021 (Model 6: O.R. = 1.245, p<0.1). These results suggest that, during the early phase of the pandemic (2020), heightened uncertainty and fear of infection were predominant. Older adults, who are perceived as a high-risk group for COVID-19, were likely to experience intensified anxiety about infection risks, particularly in areas with high population mobility. By contrast, during the escalation phase (2021), this association can be interpreted as reflecting societal adaptation mechanisms. These included improved individual compliance with preventive measures, the initiation of vaccination programs, and the transition to "living with COVID-19" policies. Over time, as the pandemic prolonged, societal adaptation to the new normal advanced, and heightened anxiety among older adults regarding population mobility appeared to diminish.

Regarding residential density, the early phase of the pandemic (2020), a positive association with depressive symptoms was observed among middle-aged adults (Model 2: O.R. = 1.207, p<0.1) and older adults (Model 3: O.R. = 1.470, p<0.05). This association became even stronger during the escalation phase (2021), with significant positive relationships observed in both middle-aged adults (Model 5: O.R. = 1.220, p<0.05) and older adults (Model 6: O.R. = 1.588, p<0.01). These results suggest that prolonged stress and anxiety regarding infection in densely populated residential areas may have contributed to increased depressive symptoms during the pandemic. Challenges in practicing effective infection

PLOS One | https://doi.org/10.1371/journal.pone.0339040  January 20, 2026

12 / 19

**Table 5. Results of COVID-19-Related Depressive Symptoms in 2021.**

| | Model 1 (Young adults) | | Model 2 (Middle-aged adults) | | Model 3 (Older adults) | |
|---|---|---|---|---|---|---|
| | O.R. | S.E. | O.R. | S.E. | O.R. | S.E. |
| **Sociodemographic Characteristics** | | | | | | |
| **Gender (1 = female)** | 1.381*** | 0.136 | 1.204** | 0.090 | 1.041 | 0.133 |
| **Age** | 1.001 | 0.017 | 1.007 | 0.005 | 0.988 | 0.011 |
| **Cohabitation Status (1 = yes)** | 0.909 | 0.125 | 0.877 | 0.102 | 1.129 | 0.199 |
| **Employment Status (1 = yes)** | 1.093 | 0.153 | 0.890 | 0.083 | 1.194 | 0.160 |
| **Education Level (1 = College Graduate)** | 1.213 | 0.181 | 0.996 | 0.089 | 0.998 | 0.150 |
| **Marital Status (1 = yes)** | 0.711** | 0.120 | 0.908 | 0.094 | 1.272 | 0.901 |
| **Apartment Residency (1 = yes)** | 0.907 | 0.092 | 0.880* | 0.063 | 0.899 | 0.103 |
| **Urban density feature** | | | | | | |
| **Population mobility ⓟ** | 1.033 | 0.092 | 0.986 | 0.070 | 1.245* | 0.157 |
| **Residential density ⓡ** | 1.110 | 0.117 | 1.220** | 0.097 | 1.588*** | 0.220 |
| **Public transportation congestion ⓣ** | 1.060 | 0.083 | 1.128** | 0.063 | 1.156 | 0.116 |
| **Social participation** | 0.990 | 0.098 | 0.995 | 0.069 | 0.616*** | 0.073 |
| **Interaction term** | | | | | | |
| **Social participation * ⓟ** | 1.038 | 0.133 | 1.032 | 0.095 | 0.875 | 0.136 |
| **Social participation * ⓡ** | 0.796* | 0.111 | 0.724*** | 0.073 | 0.515*** | 0.085 |
| **Social participation * ⓣ** | 0.867 | 0.095 | 0.866* | 0.064 | 0.750** | 0.096 |
| **N** | 1302.00 | | 2697.00 | | 1001.00 | |
| **Wald** | 31.73*** | | 42.99*** | | 46.07*** | |
| **AIC** | 5729.71 | | 11446.89 | | 4194.03 | |
| **BIC** | 5853.83 | | 11588.49 | | 4311.84 | |

O.R.=Odds Ratio, S.E.=Standard Error.

***p<0.01.

**p<0.05.

*p<0.1.

control in densely populated areas during extended periods of social distancing may have exacerbated these concerns. Policies aimed at reducing infection rates, such as encouraging people to stay at home, were associated with increased residential density and, subsequently, heightened depressive symptoms in these areas. As the pandemic continued, activities around residential areas intensified, and the increase in remote work and school closures further contributed to the rising population density in residential neighborhoods. Higher residential density implies a greater number of residents confined near their homes, potentially leading to increased depressive symptoms or stress levels. Additionally, although infection control measures primarily target workplaces, commercial hubs, and other large-scale, densely populated areas, residential areas and their surroundings often lack sufficient preventive measures. This study highlights that densely populated residential areas, coupled with stay-at-home policies, were associated with higher pandemic-related depressive symptoms owing to increased residential density and associated stressors.

Regarding public transportation congestion, in 2021, a positive association with depressive symptoms in middle-aged adults was observed (Model 5: O.R. = 1.128, p<0.01). This result may be attributed to the fact that commuting is more concentrated among middle-aged adults than among other age groups, and younger adults tend to have lower levels of anxiety regarding COVID-19. During this period, older adults generally had lower commuting and mobility rates than other age groups. In contrast, middle-aged adults not only had a higher proportion of working individuals but also a greater

likelihood of being unable to avoid commuting, which may explain the stronger association between public transportation congestion and depressive symptoms in this group. Additionally, the statistical significance of public transportation congestion on COVID-19-related depressive symptoms was low in 2020. This could be because, in 2020, most confirmed cases occurred outside the Seoul metropolitan area (e.g., Daegu and Gyeongbuk), whereas in 2021, confirmed cases were concentrated in the Seoul metropolitan area. In 2021, when large-scale outbreaks occurred in Seoul, public transportation was perceived as a potential transmission route. These findings suggest that this perception might have contributed to the observed association between public transportation congestion and COVID-19-related depressive symptoms.

Social participation showed noteworthy results as the pandemic progressed. In 2020, social participation showed a positive association with depressive symptoms among younger adults (Model 1: O.R. = 1.566, p < 0.01) and middle-aged adults (Model 2: O.R. = 1.147, p < 0.1). However, in 2021, an inverse relationship between social participation and depressive symptoms emerged among older adults (Model 6: O.R. = 0.616, p < 0.01). These findings suggest that, in 2020, social contact was associated with increased depressive symptoms related to infection anxiety. In contrast, in 2021, social contact appeared to be linked to the alleviation of depressive symptoms. This indicates that social interactions during an infectious disease outbreak can reinforce anxious perceptions of infection in 2020, contrary to traditional social capital theories. However, as society adapted to the pandemic, social contact may have served as a protective function by reducing depressive symptoms, particularly in older adults. In summary, these empirical results demonstrate that, during a pandemic such as COVID-19, the relationships between urban density characteristics, social contact, and depressive symptoms vary by age group and evolve over time. This highlights the dynamic nature of these associations in response to changing pandemic conditions.

### Results of the interaction effect of social participation

The interaction terms in Tables 4 and 5 represent the moderating effects of social participation on the relationship between urban density and COVID-19-related depressive symptoms in 2020 and 2021. The results indicate that in 2020 (Table 4), the moderating effects of social participation and urban density characteristics were particularly prominent among older adults. Among older adults, those who reported participating in social activities in areas with high population mobility had 34.5% lower odds of experiencing increased depressive symptoms than those who did not (Model 3: O.R. = 0.655, p < 0.01). Similarly, in areas with high residential density, older adults who engaged in social activities had 34.9% lower odds of experiencing heightened depressive symptoms than those who did not (Model 3: O.R. = 0.651, p < 0.01). These findings suggest that, in 2020, social participation served as an important factor in alleviating anxiety related to urban density characteristics among older adults. By contrast, for younger and middle-aged adults, the interaction effects between social participation and urban density characteristics did not have a statistically significant impact. This highlights how perceptions of the pandemic and its associated depressive symptoms vary by age group, with older adults potentially becoming more sensitive to these stressors.

The moderating effects of social participation are evident across all age groups in 2021 (Table 5). Among younger adults, the interaction between residential density and social participation demonstrated a depressive symptom mitigation effect (Model 5: O.R. = 0.796, p < 0.1). For middle-aged adults, the interaction terms of residential density (O.R. = 0.724, p < 0.01) and public transportation congestion (O.R. = 0.866, p < 0.1) with social participation also showed depressive symptom mitigation effects. The strongest moderating effect was observed among older adults. Social participation interactions with residential density (O.R. = 0.515, p < 0.01) and public transportation congestion (O.R. = 0.750, p < 0.01) significantly mitigated depressive symptoms.

These results suggest that social participation was associated with lower infection-related anxiety arising from urban density. According to previous studies, even during the depressive atmosphere of the COVID-19 pandemic, meeting with friends or engaging in social activities positively impacted mental health [42]. Furthermore, maintaining closeness in social relationships through various forms of contact has been identified as having a buffer effect, reducing the impact of

negative events [42–44]. Notably, the consistent interaction effect between residential density and social participation was significant. In 2021, this interaction was associated with reduced depressive symptoms in all age groups. Active social interactions were associated with lower depressive symptoms even in high-density environments. For example, in densely populated apartment complexes, supportive neighborhood relationships and networks can offset the stress caused by physical crowding. These findings also reveal that the mitigating effects of social participation have grown stronger over time. As the pandemic progressed, maintaining safe social interactions proved effective in reducing depressive symptoms.

## Conclusion

This study empirically analyzed the correlation between urban density characteristics and COVID-19-related depressive symptoms, and the moderating effects of social participation in 25 districts of Seoul. The main findings of this study are as follows. First, dense urban environments were generally associated with an increase in COVID-19-related depressive symptoms regardless of the pandemic phase. During the early (2020) and escalation (2021) phases of the pandemic, areas with higher population mobility, residential density, and public transportation congestion were associated with higher levels of depressive symptoms. This suggests that dense urban environments were associated with higher anxiety about infection. Second, the moderating effect of social participation was consistently observed across the pandemic phases, playing a role in reducing the increase in depressive symptoms associated with high urban density. This effect is particularly prominent in middle- and older-aged adults. Middle- and older-aged adults who engaged in social activities in areas with high residential densities had reduced odds of experiencing depressive symptoms by 27.6% and 48.5%, respectively. Third, the moderating effects of social participation and urban density on COVID-19-related depressive symptoms varied with age. Specifically, the interaction effects of the urban density variables and social participation were found to be more sensitive among older adults. This aligns with previous studies that highlighted that older adults are more vulnerable to COVID-19, anxiety about the infection, and the impact of social isolation.

Based on these findings, several practical implications for policymakers can be identified. First, urban density characteristics that contribute to mental health risks during pandemics require targeted interventions. Enhancing access to diverse green spaces—including open plazas, parks, riverside greenways, and mountain green belts characteristic of Seoul's urban landscape—can provide essential psychological relief for residents in high-density areas, as supported by research demonstrating the positive mental health effects of natural environments [32,34]. Public transportation systems should implement measures to reduce crowding and perceived infection risks, such as increased service frequency during peak hours, real-time occupancy monitoring systems, and enhanced sanitation protocols [12,36]. Urban planning policies should adopt concepts such as the "15-minute city" model that minimizes population mobility by strategically locating essential living infrastructure—including healthcare facilities, markets, and community centers—within residential neighborhoods. These coordinated environmental interventions can help mitigate the mental health impacts of urban density during infectious disease outbreaks [35–37].

Second, given the buffering effect of social participation demonstrated in this study, policies should actively facilitate safe social engagement opportunities during infectious disease outbreaks [53,54]. This includes promoting outdoor community activities, supporting neighborhood networks that maintain social bonds while minimizing infection risks, and establishing digital connectivity programs for those capable of utilizing such platforms. Furthermore, mental health policies should be phase-adaptive, evolving with pandemic progression [28–31]. During initial outbreak periods, interventions should focus on reducing acute fears through clear risk communication and accessible immediate psychological support [20,55]. As pandemics evolve into later phases, policy emphasis should shift toward supporting long-term community resilience, facilitating social reintegration programs, and actively rebuilding social networks to address the mental health deterioration resulting from prolonged distancing measures [16]. This temporal adaptation ensures that mental health support remains contextually relevant throughout different stages of infectious disease outbreaks.

Third, and particularly important, our findings reveal that older adults are more sensitive to urban density stressors and derive greater mental health benefits from social participation [56–58], indicating a critical need for age-specific policy attention. Priority interventions for older populations in dense urban areas should include establishing accessible community centers within walking distance, implementing volunteer visitor programs to reduce isolation, and creating safe outdoor spaces designed for older adult mobility and face-to-face social interaction [56,57]. When community activities are restricted due to infection control measures, supporting home-based physical activity programs can provide alternative mental health benefits for isolated older adults [59]. Healthcare and social service providers should conduct proactive outreach to identify at-risk older adults in high-density neighborhoods, particularly those living alone or with limited social networks [26,57,60]. Given that this demographic faces compounded vulnerabilities—both higher health risks from infectious diseases and greater susceptibility to urban density-related mental health impacts—targeted resources and sustained policy focus on older urban residents are essential for mitigating mental health disparities during future pandemics [56,58]. These coordinated, context-sensitive approaches can help protect the most vulnerable populations while addressing the broader mental health impacts of urban density during infectious disease outbreaks.

This study has several limitations. First, our study employed a cross-sectional design, which limits causal inference. Although we analyzed data from two points (2020 and 2021), these were independent samples rather than repeated measures of the same individuals. Therefore, our findings demonstrate associations between urban density and depression, and longitudinal studies are needed to establish causal relationships. Second, this study was conducted in 25 districts of Seoul, reflecting the city's unique urban context characterized by high population density (16,000 persons/km²) and well-developed public transportation systems. Given these distinctive characteristics differ from other cities and regions, caution is needed when generalizing our findings. Future studies should expand their scope to include a variety of cities of different sizes and characteristics to evaluate the generalizability of the results. Third, while our single-item 10-point depressive symptom scale has demonstrated validity in epidemiological research (as discussed in Methods section), it does not capture the multidimensional nature of depressive symptoms assessed by comprehensive instruments such as the PHQ-9 or CES-D. Specifically, this measure may not fully reflect the complex dimensions of depressive symptoms caused by the COVID-19 pandemic. Future studies may benefit from incorporating more detailed depression assessments to examine specific symptom domains and enable clinical severity classification. Fourth, social participation was measured as a binary variable (participation vs. no participation), which does not capture the depth, frequency, or quality of social interactions. Future research with more detailed measures of social participation—including frequency, intensity, and types of activities—would provide deeper insights into how different dimensions of social engagement buffer urban density stressors.

## Author contributions

**Data curation:** Junseong Park.

**Formal analysis:** Junseong Park.

**Funding acquisition:** Sungik Kang.

**Methodology:** Junseong Park.

**Project administration:** Ja-Hoon Koo, Sungik Kang.

**Software:** Junseong Park.

**Supervision:** Ja-Hoon Koo, Sungik Kang.

**Validation:** Sungik Kang.

**Visualization:** Junseong Park.

**Writing – original draft:** Junseong Park, Sungik Kang.

**Writing – review & editing:** Junseong Park, Ja-Hoon Koo, Sungik Kang.

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
