## [Decision Letter · Decision Letter 0]

24 Sep 2025

Dear Dr. Kang,

Thank you for submitting your manuscript to PLOS ONE. After careful consideration, we feel that it has merit but does not fully meet PLOS ONE’s publication criteria as it currently stands. Therefore, we invite you to submit a revised version of the manuscript that addresses the points raised during the review process.

The manuscript titled Urban density and depression during COVID-19 in Seoul: Age and phase differences with the moderating role of social participation addresses a highly relevant public health topic and has received generally positive reviews. The study is well-structured and provides important empirical insights into the relationship between urban density, social participation, and depressive symptoms during the pandemic. However, several essential revisions are needed before the paper can be considered for acceptance.

<h4 data-end="786" data-start="750">Required Changes for Acceptance</h4>

**Clarify Measurement of Depression**
Multiple reviewers (R1, R2, R4) raised concerns regarding the single-item 10-point depression scale. Please provide clear information on the validation of this measure (line 290) and justify its use compared to standardized instruments (e.g., PHQ-9, CES-D).

**Terminology and Clarity**

As Reviewer #2 suggests, replace vague phase terminology (“initial phase”) with precise year references (e.g., 2020 vs. 2021).Clarify whether “cumulative odds” refers specifically to proportional odds in ordinal logistic regression (R2).**Methods Transparency**  Expand details on how “mobility density” was calculated (lines 318–326, R1).Provide information on sample size, demographic characteristics, and the source of the data (R3).
**Discussion of Limitations**
Strengthen the discussion of limitations (R1, R4):• Cross-sectional design and lack of longitudinal data.• Generalizability limited to Seoul’s unique urban context.• Simplified, unidimensional measurement of depression.These should be explicitly acknowledged in the Discussion (lines 578–590).
**Interpretation of Results**
Avoid causal language (R4). Reframe results to emphasize associations rather than causality, given the cross-sectional design.**********<h4 data-end="2173" data-start="2147">Recommended Revisions</h4>
**Title**
Reviewer #1 suggests a more concise title, while Reviewer #3 finds the current title clear and appropriate. A balanced option would be to retain the essential elements but simplify phrasing, e.g., “Urban Density, Social Participation, and Depression during COVID-19 in Seoul”.
**Abstract and Introduction**
Briefly expand the methods description in the Abstract (R1).More clearly specify the literature gaps addressed by this study in the Introduction (R1).
**Discussion and Conclusion**
Add practical recommendations for policymakers based on your findings (R1, R3).Consider including multidimensional depression measures (PHQ-9, CES-D) as a recommendation for future research (R1, R4).
**Tables and Redundancy**
Consider moving Table 1 to the supplementary materials (R1).Reduce repetition in the background section (lines 127–151 and 183–204, R1).
**References**
Reviewer #1 recommends including da Cruz et al. (2022) to strengthen the discussion of social interactions and physical activity in older adults. This is appropriate and encouraged. **********<h4 data-end="3365" data-start="3328">Resolution of Reviewer Conflicts</h4>**Title** : Reviewer #1 recommends shortening the title, whereas Reviewer #3 supports the current one. My recommendation is a compromise: retain key elements (urban density, depression, COVID-19, Seoul, and social participation) while simplifying the phrasing.**Scale Justification** : Both Reviewer #2 and Reviewer #4 question the single-item depression measure, aligning with Reviewer #1’s call for validation details. This should be treated as a required revision.

We look forward to receiving your revised manuscript.

Kind regards,

Hyun Woo Jung

Academic Editor

PLOS ONE

Journal Requirements:

“This work was supported by the Korea National Research Foundation under Grant [RS-2023-00239319].”

“This work was supported by the Korea National Research Foundation under Grant [RS-2023-00239319].”

4. Please note that your Data Availability Statement is currently missing the repository name and the DOI/accession number of each dataset OR a direct link to access each database. If your manuscript is accepted for publication, you will be asked to provide these details on a very short timeline. We therefore suggest that you provide this information now, though we will not hold up the peer review process if you are unable.

Additional Editor Comments:

Reviewer #1: Thank you to the authors and the editor for the opportunity to review this study titled Urban density and depression during COVID-19 in Seoul: Age and phase differences with the moderating role of social participation. This work provides valuable empirical analysis on the effect of urban density and social participation on depressive symptoms during the COVID-19 pandemic in Seoul, addressing differences across phases and age groups.

Strengths

Relevance of the topic (lines 17–35): The article explores a highly relevant issue, delving into the link between urban density and mental health during the pandemic. This contributes significantly to the understanding of the complex interactions between urban environments and psychological well-being.

Robust methodology (lines 237–389): The use of ordered logistic regression models and the inclusion of variables such as residential density, mobility, and transportation congestion strengthen the validity of the results.

Impact of social interactions (lines 495–554): The article emphasizes the role of social participation as a moderator, highlighting effects specific to different age groups, which makes a significant contribution to the existing literature.

Suggested Modifications

Title (lines 1–6): Consider a shorter, more concise title, such as "Effects of Urban Density and Social Participation on COVID-19-Related Depression in Seoul."

Abstract (lines 17–38): Briefly expand the reference to the methodology for greater clarity.

Introduction (lines 39–109): Clearly specify gaps in the current literature and how this study addresses them.

Materials and Methods Section (lines 237–389):

Line 290: Add information on the validation of the depression measurement scale.

Lines 318–326: Provide more details about the calculation of mobility density to improve transparency.

Discussion (lines 558–596):

Line 578: Strengthen the analysis of limitations by emphasizing how the results may differ in rural contexts or cities with lower densities.

Line 590: Suggest approaches to integrate multidimensional scales for depression in future studies.

Conclusion (lines 558–596): Include practical recommendations for policymakers.

Table 1 (lines 278–279): Consider moving this table to the supplementary materials, as much of the information is already described in the text.

Repetitions (lines 127–151 and 183–204): Some paragraphs in the background section could be summarized to avoid redundancy.

Encourage the authors to include the following reference, which is particularly relevant for the role of physical activity in older adults’ mental health during the pandemic. It could be added in the Discussion section (line 560), supporting the importance of social interactions:

da Cruz, W. M., D'Oliveira, A., Dominski, F. H., Diotaiuti, P., & Andrade, A. (2022). Mental health of older people in social isolation: the role of physical activity at home during the COVID-19 pandemic. Sport Sciences for Health, 18(2), 597–602. https://doi.org/10.1007/s11332-021-00825-9

The authors have not sufficiently addressed the following limitations:

Cross-sectional nature of the data (line 578): A longitudinal design could provide greater insights into temporal changes.

Geographical contextualization (line 583): The findings may not be generalizable to contexts outside Seoul.

Unidimensional measurement of depression (line 590): Incorporating validated scales such as the PHQ-9 would allow for a more comprehensive assessment.

Reviewer #2: Good conceptulisation.

However, at certain places, refer to the timeline (2020 vs. 2021) instead of referring to it as the initial phase to enhance readability.

Are you referring to the cumulative odds? May describe precisely

Why are you using a single-item 10-point scale instead of other depression inventories?

Reviewer #3: Reviewer’s comments

Title: Urban density and depression during COVID-19 in Seoul: Age and phase differences with the moderating role of social participation

Title:

The title is clear, specific, and accurately reflects the study’s core variables (urban density, depression, COVID-19, age, phase, and social participation). It effectively signals the focus on Seoul and the moderating role of social participation.

Background and Rationale:

The introduction starts well by establishing the relevance of the topic (COVID-19’s mental health impact) and identifies clear gaps: temporal distinctions of the pandemic, age-specific differences, and the role of social participation. This gives the study a strong rationale.

Objectives:

The objective is stated clearly to analyze the relationship between depression, urban density characteristics, and the moderating effect of social participation by age and pandemic phase. This is well-aligned with the background.

Methods:

The method is briefly stated: data from 25 Seoul districts during 2020 and 2021, analyzed using ordinal logistic regression. This is appropriate, though it could briefly mention the source of the data or sample size for clarity.

Results:

Key findings are logically presented and match the objectives. The abstract clearly states (1) urban density factors are associated with increased depression, (2) social participation buffers these effects, and (3) effects vary by age especially stronger among older adults. This section is well-organized and impactful.

Conclusion/Implications:

The conclusion ties the findings back to their broader significance, highlighting the need to understand the complex interplay of urban density, social participation, and mental health. This is appropriate but could briefly mention how this understanding can inform public health or urban policy.

Strengths:

• Clear structure (background → aim → methods → results → implications)

• Relevant and timely topic

• Well-articulated findings and moderation analysis

Areas for Improvement:

• Provide a brief mention of the sample size or participant demographics.

• Clarify whether depression was measured via self-report surveys, clinical data, or another instrument.

• Include a sentence on potential policy or practical implications to strengthen the conclusion.

Overall Evaluation:

This is a well-written and coherent paper that clearly communicates the study’s purpose, methods, and main findings. With minor additions on data details and implications, it would meet high scholarly standards.

Reviewer #4: It is easy to understand from the beginning what the authors' objectives were, sounds and it is easy to read and very understandable. First, I analyzed the main points and then introduced some improvements that could give more consistency to the focused themes.

The consistent association between urban density characteristics (population mobility, residential density, and public transportation congestion) and Mental Health (increased depressive symptoms during the COVID-19 pandemic). Social participation is identified as a key factor in mitigating depressive symptoms caused by urban density. Engaging in social activities can buffer the negative mental health impacts of high-density environments, particularly among middle-aged and older adults. This finding emphasizes the protective role of social connectedness during crises.

Also reveals that the effects of urban density and social participation on depressive symptoms vary significantly by age group. Older adults are more sensitive to urban density stressors and benefit the most from social participation, while younger adults experience different dynamics. The need for age-specific mental health interventions during pandemics.

Shows also that depressive symptoms decreased overall during the escalation phase of the pandemic compared to the initial phase, suggesting societal adaptation over time. Explain how mental health evolves during prolonged crises and the importance of resilience-building measures.

The study identifies significant interaction effects between urban density characteristics and social participation, particularly in mitigating depressive symptoms in high-density environments. For example, supportive neighbourhood relationships in densely populated areas can offset stress caused by crowding. The importance of fostering social networks in urban planning and public health strategies. These ideas contribute to a deeper understanding of the complex relationships between urban density, social participation, and mental health during pandemics, offering valuable guidance for policymakers and urban planners.

Although this article is very well written and demonstrates some concerns about practical applications for future research, some methodological and coherence issues must be more specified. The study uses cross-sectional data from 2020 and 2021, which limits its ability to track changes over time for the same subjects. This restricts the ability to establish causal relationships and observe temporal dynamics. The absence of a longitudinal approach prevents the study from capturing the long-term effects of urban density and social participation on mental health as the pandemic evolved. A longitudinal design would have been more appropriate to observe how depressive symptoms evolved throughout the pandemic and to establish causal relationships between urban density, social participation, and mental health. While some results are statistically significant (e.g., odds ratios for urban density and depressive symptoms), their practical significance may be limited. For example, small changes in odds ratios may not translate into meaningful real-world impacts.

The Ordered Logistic Regression Model is suitable for analysing ordinal data, it may not fully capture the complexity of interactions between urban density, social participation, and depressive symptoms. Advanced statistical techniques, such as structural equation modelling, could have provided deeper insights into these relationships.

While the study highlights the role of urban density, it does not explore other potential factors influencing mental health, such as economic disparities, healthcare access, or cultural differences, which could provide a more comprehensive understanding.

The study focuses on 25 districts in Seoul, which has unique urban characteristics such as high population density and an extensive public transportation system. This makes it challenging to generalize the findings to smaller cities, rural areas, or urban environments with different characteristics.

Data were collected through in-person and online surveys, which may introduce biases due to differences in response rates or the willingness of participants to engage during the pandemic.

The study measures social participation as a binary variable (participation or no participation), which may not capture the depth, frequency, or quality of social interactions, potentially oversimplifying its impact on mental health.

Depressive symptoms were measured using a single-item 10-point scale. While this approach is convenient for large-scale surveys, it may not fully capture the multidimensional nature of depression, potentially oversimplifying the complex aspects of mental health during the pandemic. Using validated, multidimensional depression scales (e.g., PHQ-9 or CES-D) would have provided a more nuanced understanding of mental health impacts.

The study excludes the fifth wave (recovery phase) of the pandemic, which accounted for 95% of cumulative cases. This omission limits the understanding of how mental health evolved during the transition to normality.

The study uses cross-sectional data, which cannot establish causality assumption. However, some results are interpreted as causal relationships (e.g., urban density causing depressive symptoms), which may be misleading.

It seems that incorrect assumptions bring some methodological limitations to this article, oversimplified measurements, inadequate control of confounding variables, and potential biases in data collection and interpretation. These issues highlight the need for more rigorous and comprehensive approaches in future research.

Reviewers' comments:

Reviewer's Responses to Questions

**Comments to the Author**

1. Is the manuscript technically sound, and do the data support the conclusions?

Reviewer #1: Yes

Reviewer #2: Yes

Reviewer #3: Yes

Reviewer #4: Yes

2. Has the statistical analysis been performed appropriately and rigorously?

Reviewer #1: Yes

Reviewer #2: Yes

Reviewer #3: Yes

Reviewer #4: Yes

3. Have the authors made all data underlying the findings in their manuscript fully available?

Reviewer #1: Yes

Reviewer #2: No

Reviewer #3: Yes

Reviewer #4: Yes

4. Is the manuscript presented in an intelligible fashion and written in standard English?

Reviewer #1: Yes

Reviewer #2: Yes

Reviewer #3: No

Reviewer #4: Yes

Reviewer #1: Thank you to the authors and the editor for the opportunity to review this study titled Urban density and depression during COVID-19 in Seoul: Age and phase differences with the moderating role of social participation. This work provides valuable empirical analysis on the effect of urban density and social participation on depressive symptoms during the COVID-19 pandemic in Seoul, addressing differences across phases and age groups.

Strengths

Relevance of the topic (lines 17–35): The article explores a highly relevant issue, delving into the link between urban density and mental health during the pandemic. This contributes significantly to the understanding of the complex interactions between urban environments and psychological well-being.

Robust methodology (lines 237–389): The use of ordered logistic regression models and the inclusion of variables such as residential density, mobility, and transportation congestion strengthen the validity of the results.

Impact of social interactions (lines 495–554): The article emphasizes the role of social participation as a moderator, highlighting effects specific to different age groups, which makes a significant contribution to the existing literature.

Suggested Modifications

Title (lines 1–6): Consider a shorter, more concise title, such as "Effects of Urban Density and Social Participation on COVID-19-Related Depression in Seoul."

Abstract (lines 17–38): Briefly expand the reference to the methodology for greater clarity.

Introduction (lines 39–109): Clearly specify gaps in the current literature and how this study addresses them.

Materials and Methods Section (lines 237–389):

Line 290: Add information on the validation of the depression measurement scale.

Lines 318–326: Provide more details about the calculation of mobility density to improve transparency.

Discussion (lines 558–596):

Line 578: Strengthen the analysis of limitations by emphasizing how the results may differ in rural contexts or cities with lower densities.

Line 590: Suggest approaches to integrate multidimensional scales for depression in future studies.

Conclusion (lines 558–596): Include practical recommendations for policymakers.

Table 1 (lines 278–279): Consider moving this table to the supplementary materials, as much of the information is already described in the text.

Repetitions (lines 127–151 and 183–204): Some paragraphs in the background section could be summarized to avoid redundancy.

Encourage the authors to include the following reference, which is particularly relevant for the role of physical activity in older adults’ mental health during the pandemic. It could be added in the Discussion section (line 560), supporting the importance of social interactions:

da Cruz, W. M., D'Oliveira, A., Dominski, F. H., Diotaiuti, P., & Andrade, A. (2022). Mental health of older people in social isolation: the role of physical activity at home during the COVID-19 pandemic. Sport Sciences for Health, 18(2), 597–602. https://doi.org/10.1007/s11332-021-00825-9

The authors have not sufficiently addressed the following limitations:

Cross-sectional nature of the data (line 578): A longitudinal design could provide greater insights into temporal changes.

Geographical contextualization (line 583): The findings may not be generalizable to contexts outside Seoul.

Unidimensional measurement of depression (line 590): Incorporating validated scales such as the PHQ-9 would allow for a more comprehensive assessment.

Reviewer #2: Good conceptulisation.

However, at certain places, refer to the timeline (2020 vs. 2021) instead of referring to it as the initial phase to enhance readability.

Are you referring to the cumulative odds? May describe precisely

Why are you using a single-item 10-point scale instead of other depression inventories?

Reviewer #3: Reviewer’s comments

Title: Urban density and depression during COVID-19 in Seoul: Age and phase differences with the moderating role of social participation

Title:

The title is clear, specific, and accurately reflects the study’s core variables (urban density, depression, COVID-19, age, phase, and social participation). It effectively signals the focus on Seoul and the moderating role of social participation.

Background and Rationale:

The introduction starts well by establishing the relevance of the topic (COVID-19’s mental health impact) and identifies clear gaps: temporal distinctions of the pandemic, age-specific differences, and the role of social participation. This gives the study a strong rationale.

Objectives:

The objective is stated clearly to analyze the relationship between depression, urban density characteristics, and the moderating effect of social participation by age and pandemic phase. This is well-aligned with the background.

Methods:

The method is briefly stated: data from 25 Seoul districts during 2020 and 2021, analyzed using ordinal logistic regression. This is appropriate, though it could briefly mention the source of the data or sample size for clarity.

Results:

Key findings are logically presented and match the objectives. The abstract clearly states (1) urban density factors are associated with increased depression, (2) social participation buffers these effects, and (3) effects vary by age especially stronger among older adults. This section is well-organized and impactful.

Conclusion/Implications:

The conclusion ties the findings back to their broader significance, highlighting the need to understand the complex interplay of urban density, social participation, and mental health. This is appropriate but could briefly mention how this understanding can inform public health or urban policy.

Strengths:

• Clear structure (background → aim → methods → results → implications)

• Relevant and timely topic

• Well-articulated findings and moderation analysis

Areas for Improvement:

• Provide a brief mention of the sample size or participant demographics.

• Clarify whether depression was measured via self-report surveys, clinical data, or another instrument.

• Include a sentence on potential policy or practical implications to strengthen the conclusion.

Overall Evaluation:

This is a well-written and coherent paper that clearly communicates the study’s purpose, methods, and main findings. With minor additions on data details and implications, it would meet high scholarly standards.

Reviewer #4: It is easy to understand from the beginning what the authors' objectives were, sounds and it is easy to read and very understandable. First, I analyzed the main points and then introduced some improvements that could give more consistency to the focused themes.

The consistent association between urban density characteristics (population mobility, residential density, and public transportation congestion) and Mental Health (increased depressive symptoms during the COVID-19 pandemic). Social participation is identified as a key factor in mitigating depressive symptoms caused by urban density. Engaging in social activities can buffer the negative mental health impacts of high-density environments, particularly among middle-aged and older adults. This finding emphasizes the protective role of social connectedness during crises.

Also reveals that the effects of urban density and social participation on depressive symptoms vary significantly by age group. Older adults are more sensitive to urban density stressors and benefit the most from social participation, while younger adults experience different dynamics. The need for age-specific mental health interventions during pandemics.

Shows also that depressive symptoms decreased overall during the escalation phase of the pandemic compared to the initial phase, suggesting societal adaptation over time. Explain how mental health evolves during prolonged crises and the importance of resilience-building measures.

The study identifies significant interaction effects between urban density characteristics and social participation, particularly in mitigating depressive symptoms in high-density environments. For example, supportive neighbourhood relationships in densely populated areas can offset stress caused by crowding. The importance of fostering social networks in urban planning and public health strategies. These ideas contribute to a deeper understanding of the complex relationships between urban density, social participation, and mental health during pandemics, offering valuable guidance for policymakers and urban planners.

Although this article is very well written and demonstrates some concerns about practical applications for future research, some methodological and coherence issues must be more specified. The study uses cross-sectional data from 2020 and 2021, which limits its ability to track changes over time for the same subjects. This restricts the ability to establish causal relationships and observe temporal dynamics. The absence of a longitudinal approach prevents the study from capturing the long-term effects of urban density and social participation on mental health as the pandemic evolved. A longitudinal design would have been more appropriate to observe how depressive symptoms evolved throughout the pandemic and to establish causal relationships between urban density, social participation, and mental health. While some results are statistically significant (e.g., odds ratios for urban density and depressive symptoms), their practical significance may be limited. For example, small changes in odds ratios may not translate into meaningful real-world impacts.

The Ordered Logistic Regression Model is suitable for analysing ordinal data, it may not fully capture the complexity of interactions between urban density, social participation, and depressive symptoms. Advanced statistical techniques, such as structural equation modelling, could have provided deeper insights into these relationships.

While the study highlights the role of urban density, it does not explore other potential factors influencing mental health, such as economic disparities, healthcare access, or cultural differences, which could provide a more comprehensive understanding.

The study focuses on 25 districts in Seoul, which has unique urban characteristics such as high population density and an extensive public transportation system. This makes it challenging to generalize the findings to smaller cities, rural areas, or urban environments with different characteristics.

Data were collected through in-person and online surveys, which may introduce biases due to differences in response rates or the willingness of participants to engage during the pandemic.

The study measures social participation as a binary variable (participation or no participation), which may not capture the depth, frequency, or quality of social interactions, potentially oversimplifying its impact on mental health.

Depressive symptoms were measured using a single-item 10-point scale. While this approach is convenient for large-scale surveys, it may not fully capture the multidimensional nature of depression, potentially oversimplifying the complex aspects of mental health during the pandemic. Using validated, multidimensional depression scales (e.g., PHQ-9 or CES-D) would have provided a more nuanced understanding of mental health impacts.

The study excludes the fifth wave (recovery phase) of the pandemic, which accounted for 95% of cumulative cases. This omission limits the understanding of how mental health evolved during the transition to normality.

The study uses cross-sectional data, which cannot establish causality assumption. However, some results are interpreted as causal relationships (e.g., urban density causing depressive symptoms), which may be misleading.

It seems that incorrect assumptions bring some methodological limitations to this article, oversimplified measurements, inadequate control of confounding variables, and potential biases in data collection and interpretation. These issues highlight the need for more rigorous and comprehensive approaches in future research.

**Do you want your identity to be public for this peer review?** For information about this choice, including consent withdrawal, please see our Privacy Policy

Reviewer #1: **Yes**:  Pierluigi Diotaiuti

Reviewer #2: No

Reviewer #3: No

Reviewer #4: No

---

## [Author Response · Author response to Decision Letter 1]

7 Nov 2025

Required Changes for Acceptance

RC_1. Clarify Measurement of Depression

Reviewer's Comment:

Multiple reviewers (R1, R2, R4) raised concerns regarding the single-item 10-point depression scale. Please provide clear information on the validation of this measure (line 290) and justify its use compared to standardized instruments (e.g., PHQ-9, CES-D).

Response:

Thank you for this important comment. We have now added validation evidence and justification for the single-item measure.

1. Validation Evidence Added (Line 290)

We added a paragraph citing empirical studies demonstrating validity of single-item depression measures in epidemiological research:

• Zimmerman et al. (2006): Strong correlation with 18-item CUDOS (r=0.78, p<0.001) and clinician-rated CGI-S (r=0.64, p<0.001) among 562 outpatients

• van Rijsbergen et al. (2014): 94% discriminative accuracy (AUC=0.94) against structured clinical interviews (SCID-I)

• Mallon et al. (2002): 81% agreement with 14-item HADS in large epidemiological study

2. Pandemic-Specific Justification Added (Line 295)

We explained why this approach was appropriate for COVID-19 surveillance:

• Rapid assessment: Essential for timely public health response during evolving pandemic

• Large-scale feasibility: Minimized respondent burden (N=5,000 per wave)

• Pandemic-specific focus: Our measure explicitly assesses depressive symptoms "due to the pandemic," differentiating pandemic-attributed depression from general depression

• International precedent: Khajuria et al. (2021) successfully used single-item assessment among 2,527 healthcare workers across 41 countries during COVID-19

3. Distinction from PHQ-9/CES-D

Our measure serves different objectives than clinical instruments:

• PHQ-9/CES-D: Clinical diagnosis and comprehensive symptom assessment

• Our measure: Population-level surveillance of pandemic-attributed mental health impacts

This design aligns with municipal health monitoring objectives prioritizing rapid, large-scale assessment over detailed clinical evaluation.

4. Limitations Acknowledged

We revised the existing limitation statement to acknowledge measurement constraints while referencing the validation evidence:

Original: "Third, the accuracy of measuring COVID-19-related depressive symptoms was limited. In this study, depressive symptoms were measured using a single-item 10-point scale..."

Revised: "Third, while our single-item 10-point depression scale has demonstrated validity in epidemiological research (as discussed in Methods section), it does not capture the multidimensional nature of depression assessed by comprehensive instruments such as the PHQ-9 or CES-D. Specifically, this measure may not fully reflect the complex dimensions of depressive symptoms caused by the COVID-19 pandemic. Future studies may benefit from incorporating more detailed depression assessments to examine specific symptom domains and enable clinical severity classification."

Terminology and Clarity

RC_2. As Reviewer #2 suggests, replace vague phase terminology (“initial phase”) with precise year references (e.g., 2020 vs. 2021).

Response:

Thank you for this valuable suggestion to enhance clarity. We have systematically revised the entire manuscript to replace vague phase terminology with precise year references.

1. Global Terminology Revision

All vague phase terminology has been replaced with year-specific references:

• "initial phase" → "2020"

• "escalation phase" → "2021"

• "early pandemic" → "2020"

• "pandemic escalation" → "2021"

2. Key Section Revisions

Materials and Methods (Lines 239-280):

• Original: "...dividing it into two main phases: the initial and escalation phases"

• Revised: "...dividing it into two periods: 2020 and 2021"

Results Section Headings:

• "Results in the Initial Phase" → "Results in 2020"

• "Results in the Escalation Phase" → "Results in 2021"

3. Updated Table Titles

• Table 4: "Results of COVID-19-Related Depressive Symptoms in 2020"

• Table 5: "Results of COVID-19-Related Depressive Symptoms in 2021"

4. Contextual Phase References

Where analytical context requires phase references, we use parenthetical notation:

• Example: "During 2020 (initial phase of the pandemic)"

• This maintains clarity while preserving the analytical framework

RC_3. Clarify whether “cumulative odds” refers specifically to proportional odds in ordinal logistic regression (R2).

Response:

Thank you for requesting clarification on the terminology. Yes, we are referring to cumulative odds in the context of the proportional odds model.

In the ordered logistic regression analysis used in this study:

- We calculate cumulative odds, defined as P(Y ≤ j) / P(Y > j), which represents the ratio of the probability of being at or below a particular category versus being above that category for each threshold j

- The model operates under the proportional odds assumption, meaning that the effect coefficients (β) remain constant across all cumulative logits, while only the threshold parameters (intercepts) vary across categories

We have revised the manuscript to clarify this terminology. Specifically, we have:

1. Explicitly noted that the ordered logistic model is a proportional odds model

2. Added a precise definition of cumulative odds with the mathematical expression P(Y ≤ j) / P(Y > j)

3. Explained the proportional odds assumption underlying the model

These revisions ensure that readers understand both the specific type of odds being modeled (cumulative odds) and the key assumption (proportional odds) that characterizes this analytical approach.

Methods Transparency

RC_4. Expand details on how “mobility density” was calculated (lines 318–326, R1).

Response:

Thank you for this important request for methodological transparency. We have expanded the description of urban density measures, including population mobility, in the Methods section.

Mobility Density Measurement

Mobility density was calculated using Seoul's Living Population dataset, which provides daily aggregated mobile population counts at the district level. This dataset is publicly available through the Seoul Metropolitan Government's open data platform. The measure represents the daily maximum mobile population in each district divided by the district's geographic area (km²), expressed as persons per square kilometer.

Data Source and Collection Methodology

The Seoul Living Population dataset is derived from KT (Korea Telecom) LTE network data, which captures real-time location patterns through mobile base station signals. The system tracks population movements on an hourly basis, distinguishing between nighttime residential locations and daytime activity locations (e.g., workplaces, commercial areas).

Residence Classification Algorithm

An individual's residence is algorithmically determined using the following criteria:

• Continuous presence in a location for at least 4 hours between midnight and 6:00 AM

• Consistent repetition of this pattern over a minimum 14-day observation period

• Coverage includes Korean nationals and long-term foreign residents with active mobile subscriptions

Rationale for Using Daily Maximum Values

Rather than daily averages, we used the daily maximum mobile population—the highest hourly count recorded each day. This approach offers three key advantages:

1. Peak exposure assessment: Captures moments of highest infection transmission risk during rush hours, lunch periods, and other times of concentrated human contact

2. Infrastructure stress points: Reflects maximum congestion levels in public transportation and shared spaces where disease transmission is most likely

3. Epidemiological validity: Prior research demonstrates that peak mobility volumes are stronger predictors of infectious disease transmission than average daily movements

Calculation:

Mobility Density = Daily Maximum Mobile Population (persons) / District Area (km²)

RC_5. Provide information on sample size, demographic characteristics, and the source of the data (R3).

Response:

We have enhanced the manuscript with detailed information about sample size, demographics, and data sources.

1. Sample Size Clarification

• Original: " The Seoul Citizen Survey targeted 5,000 individuals aged 15 years and older residing in Seoul..."

• Revised: " The Seoul Citizen Survey annually targeted 5,000 individuals aged 15 years and older residing in Seoul, yielding a total sample of 10,000 participants across the two study periods (2020: n=5,000; 2021: n=5,000)."

2. Demographic Characteristics

• Original: "These characteristics included sex (male, female), age, educational level..."

• Revised: " Participants were categorized into three age groups for analysis: young adults (19-34 years), middle-aged adults (35-64 years), and older adults (65 years and above), with sample distributions detailed in Table 2. These characteristics included sex (male, female), age, educational level..."

3. Table 2 Enhancement

• Original: "Table 2 illustrates the differences in COVID-19-related depressive symptoms according to year and age group."

• Revised: Table 2 presents the sample distribution and differences in COVID-19-related depressive symptoms by pandemic phase and age group. In 2020, the sample comprised 1,227 young adults (19-34 years), 2,695 middle-aged adults (35-64 years), and 1,080 older adults (≥65 years). In 2021, the distribution was 1,302 young adults, 2,697 middle-aged adults, and 1,001 older adults.

RC_6. Discussion of Limitations

Strengthen the discussion of limitations (R1, R4):

• Cross-sectional design and lack of longitudinal data.

• Generalizability limited to Seoul’s unique urban context.

• Simplified, unidimensional measurement of depression.

These should be explicitly acknowledged in the Discussion (lines 578–590).

Response:

Thank you for this critical feedback. We have substantially strengthened our limitations discussion to explicitly acknowledge these methodological constraints with the following key enhancements:

1. Cross-sectional Design Limitation

Clarified that our two-timepoint data represent independent samples rather than repeated measures, explicitly stating this limits causal inference and that our findings demonstrate associations, not causality. Added specific recommendation for longitudinal studies to establish causal relationships.

2. Limited Generalizability

Added specific contextual details characterizing Seoul's unique urban environment (population density of 16,000 persons/km²; extensive public transportation system) to help readers understand why generalization to other contexts requires caution. Recommended future research across diverse urban settings.

3. Single-item Depression Measure Limitation

Reframed from vague "accuracy limitation" to explicitly acknowledge the measure does not capture depression's multidimensional nature. Distinguished our epidemiological surveillance objective from clinical assessment, while recommending comprehensive instruments (PHQ-9, CES-D) for future research to examine specific symptom domains and severity classification.

These revisions transform our original brief limitation statements into a comprehensive discussion that transparently acknowledges methodological constraints while providing concrete directions for future research.

RC_7. Interpretation of Results

Avoid causal language (R4). Reframe results to emphasize associations rather than causality, given the cross-sectional design.

Response:

Thank you for this crucial methodological point. We have systematically revised the entire manuscript to remove causal language and emphasize associations consistent with our cross-sectional design.

1. Abstract Revisions (Lines 33-40)

First finding:

• Original: "...were consistently associated with increased depression throughout the pandemic"

• Revised: "...were consistently associated with higher levels of depression across both study periods (2020 and 2021)"

Second finding:

• Original: " social participation demonstrated a buffering effect, mitigating the depressive impact of urban density characteristics regardless of the pandemic phase."

• Revised: " Second, social participation showed a significant moderating association with the relationship between urban density characteristics and depression, regardless of the pandemic "

2. Results and Discussion Revisions

Temporal patterns:

• Original: " This finding suggests the possibility of adapting to the pandemic over time. "

• Revised: " This finding indicates a temporal decline in COVID-19-related depressive symptoms during the study period. "

Policy associations:

• Original: " ..., such as encouraging people to stay at home, likely led to increased residential density "

• Revised: " ..., such as encouraging people to stay at home, were associated with increased residential density "

• Original: " coupled with stay-at-home policies, can amplify pandemic-related depressive symptoms. "

• Revised: " coupled with stay-at-home policies, were associated with higher pandemic-related depressive symptoms "

• Original: "These results suggest that social participation mitigates infection-related anxiety arising from urban density."

• Revised: "These results suggest that social participation was associated with lower infection-related anxiety arising from urban density."

• Original: "Active social interactions appear to alleviate depressive symptoms even in high-density environments."

• Revised: "Active social interactions were associated with lower depressive symptoms even in high-density environments."

4. Conclusion Revisions

• Original: "This suggests that dense urban environments can amplify anxiety about infections."

• Revised: "This suggests that dense urban environments were associated with higher anxiety about infections."

• Original: "playing a role in reducing the increase in depressive symptoms caused by high urban density"

• Revised: "playing a role in reducing the increase in depressive symptoms associated with high urban density"

Recommended Revisions

RR_1. Title

Reviewer #1 suggests a more concise title, while Reviewer #3 finds the current title clear and appropriate. A balanced option would be to retain the essential elements but simplify phrasing, e.g., “Urban Density, Social Participation, and Depression during COVID-19 in Seoul”.

Response:

Thank you for the valuable feedback regarding the manuscript title. We have carefully considered both reviewers' comments and your recommendation for a compromise approach. We have revised the title to: "Urban density and depression during COVID-19 in Seoul: Moderating effects of social participation"

This revision maintains all key elements (urban density, depression, COVID-19, Seoul, and social participation) while achieving greater conciseness by removing detailed methodological aspects (age and phase differences). These important distinctions remain thoroughly addressed in the abstract and main text.

We believe this balanced approach satisfies both the call for brevity and the need for clarity in communicating our study's core contribution.

Abstract and Introduction

RR_2. Briefly expand the methods description in the Abstract (R1).

Response:

We appreciate this valuable suggestion. We have expanded the methods description in the Abstract to provide greater clarity and transparency about our study design and analytical approach. The revised Abstract now includes:

1) The measurement method for depressive symptoms (10-point self-report scale)

2) Specific urban density variables examined (population mobility density, residential density, and public transportation congestion)

3) Data sources for these variables (mobile phone signal data and building registries)

4) The analytical strategy of using separate models for three age groups

These additions provide readers with a clearer understanding of our methodological approach without making the Abstract overly technical or exceeding the 300-word limit required by PLOS ONE.

RR_3. More clearly specify the literature gaps addressed by this study in the Introduction (R1).

Response:

We sincerely thank the reviewer for this critical suggestion. We acknowledge that while our original manuscript did identify research gaps in the Literature Review

---

## [Editor Report · Decision Letter 1]

1 Dec 2025

Urban density and depression during COVID-19 in Seoul: Moderating effects of social participation

PONE-D-25-01974R1

Dear Dr. Kang,

We’re pleased to inform you that your manuscript has been judged scientifically suitable for publication and will be formally accepted for publication once it meets all outstanding technical requirements.

Kind regards,

Hyun Woo Jung

Academic Editor

PLOS ONE
---

## [Editor Report · Acceptance letter]

PONE-D-25-01974R1

PLOS One

Dear Dr. Kang,

I'm pleased to inform you that your manuscript has been deemed suitable for publication in PLOS One. Congratulations! Your manuscript is now being handed over to our production team.

Kind regards,

on behalf of

Dr. Hyun Woo Jung

Academic Editor

PLOS One